# The cost of oral cancer: A systematic review

**Rejane Faria Ribeiro-Rotta**[1☯]*, **Eduardo Antônio Rosa**[1☯], **Vanessa Milani**[1☯], **Nadielle Rodrigues Dias**[1☯], **Danielle Masterson**[2‡], **Everton Nunes da Silva**[3‡], **Ana Laura de Sene Amâncio Zara**[1☯]

**1** School of Dentistry, Universidade Federal de Goiás (UFG), Goiânia, Goiás, Brazil, **2** Universidade Federal do Rio de Janeiro, Rio de Janeiro, Brazil, **3** Universidade de Brasília (UnB), Brasília, Federal District, Brazil

☯ These authors contributed equally to this work.
‡ DM and ENS also contributed equally to this work.
* rejanefrr@ufg.br

**Data Availability Statement:** All data files are available from the Figshare database (accession number https://figshare.com/s/ f7eb4990efeb5021f131).

## Abstract

Although clinical and epidemiological aspects of oral cancers (OC) are well-documented in the literature, there is a lack of evidence on the economic burden of OC. This study aims to provide a comprehensive systematic assessment on the economic burden of OC based on available evidence worldwide. A systematic review was conducted. The population was any individual, who were exposed to OC, considered here as lip (LC), oral cavity (OCC), or oro-pharynx (OPC) cancer. The outcome was information on direct (medical and non-medical) and indirect (productivity loss and early death) costs. The data sources included Scopus, Web of Science, Cochrane, BVS, and NHS EED. A search of grey literature (ISPOR and INAHTA proceedings) and a manual search in the reference lists of the included publications were performed (PROSPERO no. CRD42020172471). We identified 24 studies from 2001 to 2021, distributed by 15 countries, in 4 continents. In some developed western countries, the costs of LC, OCC, and OPC reached an average of Gross Domestic Product per capita of 18%, 75%, and 127%, respectively. Inpatient costs for OC and LC were 968% and 384% higher than those for outpatients, respectively. Advanced cancer staging was more costly (from ~22% to 373%) than the early cancer staging. The economic burden of oral cancer is substantial, though underestimated.

## Introduction

Detection of oral cancer does not demand elaborate screening tests such as breast, prostate, and colon cancers. Oral cancer can be easily and effectively detected early with oral inspection during routine dental consultations and integrated in primary care [1]. To achieve this goal, current efforts must include target programs to educate high-risk persons and primary care providers about the main aspects of early detection [2]. Oral cancer staging plays an important role in survival rate, with early-stage (I and II) and advanced-stage (III and IV) lesions having a 5-year survival rate of 80% and 50% or less, respectively [3]. Additionally, advanced stages require more aggressive combined interventions, and consequently more expensive treatments. There are also equity concerns about oral cancers, since they asymmetrically affect

**Funding:** The authors received no specific funding for this work.

**Competing interests:** The authors have declared that no competing interests exist.

different population groups and countries. Older, heavier male users of tobacco and alcohol, and people from low socioeconomic strata, as well as those who have a poor dietary intake are populations who are at a high risk of developing oral cancer [4]. Regarding geographical locations, the highest incidence rates occur in three low- and middle-income countries (Pakistan, Brazil, and India) [4]. There is also a growing incidence of oral and oropharynx cancer among young patients (<45 years), particularly in Africa, the Middle East, and Asia [5].

Although clinical and epidemiological aspects of oral cancers are well-documented in the literature, there is a lack of evidence on the economic burden of oral cancers worldwide. Cost-of-illness studies can provide information on the monetary consequences of a disease or condition, including healthcare costs and productivity losses, and its impact on societal or public health expenditure [6]. This information can be used to estimate avoidable costs if policies/programmes are implemented to reduce the prevalence of this disease. When available, it also can inform costs stratified by stages of the disease. In the United Kingdom, average treatment cost for oral cancer can range from I$ 3,343 in the early stages to I$24,890 in the advanced stages [7]. Cost-of-illness can also be used to inform priority setting, by providing estimates of how big a problem is in terms of costs [8]. Moreover, gathering information on costs may encourage decision makers to implement strategies for detecting and screening populations at high-risk of developing oral cancer, particularly by comparing costs at different stages of the disease. To the best of our knowledge, up to now there are no systematic reviews that synthesize evidence on the economic burden of oral cancer. The objective of this study is to provide a comprehensive systematic assessment of the economic burden of oral cancer based on available evidence worldwide.

## Methods

A systematic review of studies revealing the costs of lip cancer (LC), oral cavity cancer (OCC), and oropharyngeal cancer (OPC) was conducted, taking into account any cost perspective (societal, third-party players, public systems). The method used was guided by the concepts of the Joanna Briggs Institute (JBI) [9] and in accordance with the Preferred Reporting Items for Systematic Reviews and Meta-Analysis (PRISMA) guidelines [10]. A systematic review protocol can be found as a preprint on Research Square (https://www.researchsquare.com/article/rs-34637/v1). This protocol was reformulated, and the final version can be found in Prospero (CRD42020172471).

### Problem specification

What is the economic burden of oral cancer, including direct and indirect costs?

The question was framed using the acronym PEO (Population, Exposure, Outcome), which was used to define the search strategy. The population (P) considered for publication searching was any individual (human) or groups of individuals, without restriction of age, sex, race, or socioeconomic status, who were exposed (E) to oral cancer, considered here as LC, OCC, or OPC. The outcome (O) required from the publications was information on direct (medical and non-medical) and indirect (productivity loss and early death) costs.

### Eligibility criteria

Original studies on the cost of oral cancer, which included direct and/or indirect costs, or that provided estimates per patient (average cost or by clinical stage) or economic burden as percentage of GDP or national healthcare expenditure were included in the review. No language or year of publication restriction was established.

Publications that met the following criteria were excluded:

- Types of study such as: editorial, letters to the editor, systematic and non-systematic reviews of the literature, meta-analyses, case reports, case series, clinical trials.

- Studies that estimated specific item components of oral cancer cost (e.g., only surgery or medication, etc).

- Studies that addressed specific analyses, such as cost-effectiveness, cost-utility, cost-benefit, cost-minimization.

## Information sources

A systematic literature search was carried out through a comprehensive search of databases in PubMed, Scopus, Web of Science, BVS (Biblioteca Virtual em Saúde) and NHS Economic Evaluation Database up to March 31, 2021. We also manually searched the references of the articles included for additional studies. Additionally, our search was supplemented by gray literature, with the search of abstracts of conference proceedings from annual meetings of the following societies: International Society for Pharmacoeconomics and Outcomes Research (ISPOR) and International Network of Agencies for Health Technology Assessment (INAHTA) [11] (accessed March 31, 2021).

## Design of search strategy

To identify relevant cost-of-illness studies for LC, OCC, and OPC, appropriate disease-related MeSH terms were used (Additional file 1, available via https://figshare.com/s/f7eb4990efeb5021f131). To determine the search strategy, descriptors were selected by building a table (concept mapping). The table rows were allocated for each item of the acronym PEO and the columns for PubMed controlled vocabulary terms (Medical Subject Headings–MeSH), their subcategories (entry terms; see also), and uncontrolled vocabulary (free terms) usually obtained from titles and abstracts of the main publications, books, and gray literature on the research theme. After the PubMed MeSH controlled vocabulary tree was explored, terms were tested in the PubMed database and the most relevant descriptors were selected, and a search strategy was built (Additional file 2, available via https://figshare.com/s/f7eb4990efeb5021f131). The search strategy defined for PubMed was adapted for searches in the other databases.

All publications identified in the databases were exported to the Mendeley Reference Manager (Mendeley®, Elsevier, version 1.19.5/20019) for duplicate removal. After that, all publications were exported to Rayyan® software (Rayyan QCRI, Qatar Computing Research Institute–Data Analytics) [12] for the selection process.

## Selection process

The stages of the selection process included at least one reviewer from each of the following fields of knowledge: oral cancer (EAR; VM; NRD; RFRR); epidemiology (ALSAZ); and / or health economics (ENS). Three reviewers (EAR; VM; NRD) read the title and abstract of publications using the software Rayyan (Rayyan QCRI). Kappa statistic was calculated to assess agreement between reviewers, in pairs, in the eligibility stage, with a significance level of 5% (p<0.05). The scale of Kappa value interpretation was as following: <0 no agreement; 0–0.20 slight; 0.21–0.40 fair; 0.41–0.60 moderate; 0.61–0.80 substantial and 0.81–1.0 perfect. All studies identified were screened based on the eligibility criteria and were forwarded for full-text review. Contact with the authors was established for the screened studies not available in full

text. Two reviewers (EAR; ENS) independently read the full text for inclusion. Additional reviewers (RFRR; ALSAZ) were consulted for consensus in case of disagreement between the first two (EAR; ENS). Reviewers underwent training prior to the publication selection process, which was performed using 100 screened publications.

## Data collection

An instrument was built to extract the relevant data on cost methodologies, designs, and approaches, using Research Electronic Data Capture (REDCap) [13]. This instrument included the following variables:

a. Study identification: first author; country; journal and year of publication.

b. Main study design characteristics: type of study (cost-of-illness study or another type of study that provides cost-of-illness information of oral cancer); epidemiological approach (longitudinal or cross-sectional or case control); sample (number, age, type of cancer, cancer anatomical site and stage); retrospective or prospective data gathering; data source; perspective of the analysis (societal, government, health insurance provider, hospital); time horizon; presence of a control group (patients not affected by oral cancer); location/setting (country, state, or city); cost-of-illness based approach (prevalence-based or incidence-based); estimation of resources and costs (single study-based or model-based); assumptions adopted (structural or other assumptions underpinning the study); year of cost estimation; currency; sensitivity analysis; use of discount rate; funding sources; conversion; data source (primary or secondary database). The perspective of studies was defined as: i) societal, which includes direct and indirect costs and/or out-of-pocket costs from patient point of view; ii) government (public payer), includes direct costs only; iii) health insurance provider (private payer), which includes direct costs reimbursed by the private health insurers; and iv) hospital, which includes direct cost charged by just one hospital, unless the authors explicitly reported the government or health insurer perspectives.

c. Type of cost estimated: direct healthcare costs (hospitalization, surgery, chemotherapy, radiotherapy, intensive care unit, emergency room, physical therapists, speech therapists, medication, laboratory tests, imaging diagnosis and follow-up); direct non-healthcare costs (social services and transportation costs), indirect costs (productivity loss, early death).

d. Primary study outcomes: costs related to oral cancer in patients, reported in monetary units or economic burden as a percentage of Gross Domestic Product (GDP) or national healthcare expenditure.

e. Additional outcome: if the studies provided a specific breakdown of costs, this information was reported as a secondary outcome (outpatient and inpatient costs; cost by clinical stage; primary and recurrent tumor cost). We also calculated the economic burden of OC at individual level, by dividing the OC costs per patient by the GDP per capita of the country under investigation. This measure would indicate how catastrophic those costs could be for an average citizen (GDP per capita).

Data extraction was carried out by at least two of four reviewers (ALSAZ; EAR; ENS; RFRR), in a double-blind process, and disagreements were decided by consensus.

## Data synthesis

All studies meeting the eligibility criteria were included in the study and critically appraised using the Larg & Moss's guide [14] for assessing cost-of-illness. This checklist includes three

domains: analytical framework; methodology and data; analysis and reporting. The method for assessing quality of individual studies was done at both the outcome and study level, independently, and in duplicate (EAR, ENS), and discrepancies were resolved by consensus. We provided a global score for the quality of each study by calculating the total number of points rated as "yes" and "not applicable (NA)". Percentage intervals were established for meeting the items of the quality assessment instrument applied to the included studies: >80%; between 79% and 50%; and less than 50%. The average and standard deviation (SD) of the scores were calculated. The average of the scores were compared between study design groups (longitudinal studies, cross-sectional and case control studies, and cross-sectional studies based on information system data) and by domains, using a one-way analysis of variance (ANOVA) (p<0.05), by Open-Source Epidemiologic Statistics for Public Health (OpenEpi), version 3.01 [15]. In this section, this was considered the risk of bias information obtained from each study.

To calculate the percentage of the burden of the cost of oral cancer, GDP per capita of the countries where the studies were carried out was considered and converted to International Dollars (I$) by Purchasing Power Parity—PPP (2019) [16].

The results were presented in narrative form, using the Synthesis Without Meta-Analysis (SWiM) reporting guideline [17], and the main results were presented in tables.

## Results

The search procedure is shown in the PRISMA flow diagram [10] (Fig 1). The systematic literature search identified 12,391 potentially relevant articles. After removal of duplicates, 6,864 studies were screened for inclusion (Fig 1). Following title and abstract review, full-text articles were assessed (n = 44) and excluded (n = 20) for the following reasons—they were not an oral cancer cost study, did not include a specific intervention cost, there was a head and neck cancer cost study that did not present oral cancer cost separately, only proceedings available and there were no abstracts. The author or co-authors were contacted by email for the nine studies which were unobtainable, and for which only the abstracts or the title was available. There was only one answer from all of these authors, stating that they had not published the full study. Overall, 24 studies met all the eligibility criteria and were included in the systematic review.

In the eligibility stage, Kappa coefficient was 0.83 (perfect) between the EAS and VM reviewers; 0.78 (substantial) between EAS and NRD and 0.78 (substantial) between VM and NRD.

### Study characteristics

The study characteristics are summarized in Tables 1 and 2. The studies identified were published from 2001 to 2021 and distributed by continent as follows: Europe (n = 9), Asia (n = 7), America (n = 6), Oceania (n = 1), and 1 global study stratified by region and income of 195 countries. The study population size ranged from a minimum of 69 (Sri Lanka) [18] to a maximum of 62,265 (Korea) [19]. Four studies [20–22] estimated costs by procedures and not per individual. The studies included investigated a wide variety of anatomical sites of the head and neck region, using a non-standardized terminology to identify them (Table 1).

The 24 included studies were stratified by study design, which revealed five longitudinal studies [23–27] with a time horizon varying from one to five years. Of the 18 cross-sectional studies, 10 were cost-of-illness [18–22, 28–32], 8 cost analysis [33–40] and one was a case control study [41]. The 24 studies used primary and/or secondary data sources, of which four were based on information system data [20–22, 28]. The most frequent perspective of the studies was hospital (n = 7) [21, 32, 33, 35, 37–39]. Two studies used estimation of resources and cost based on mathematical models [31, 35] (Table 2).

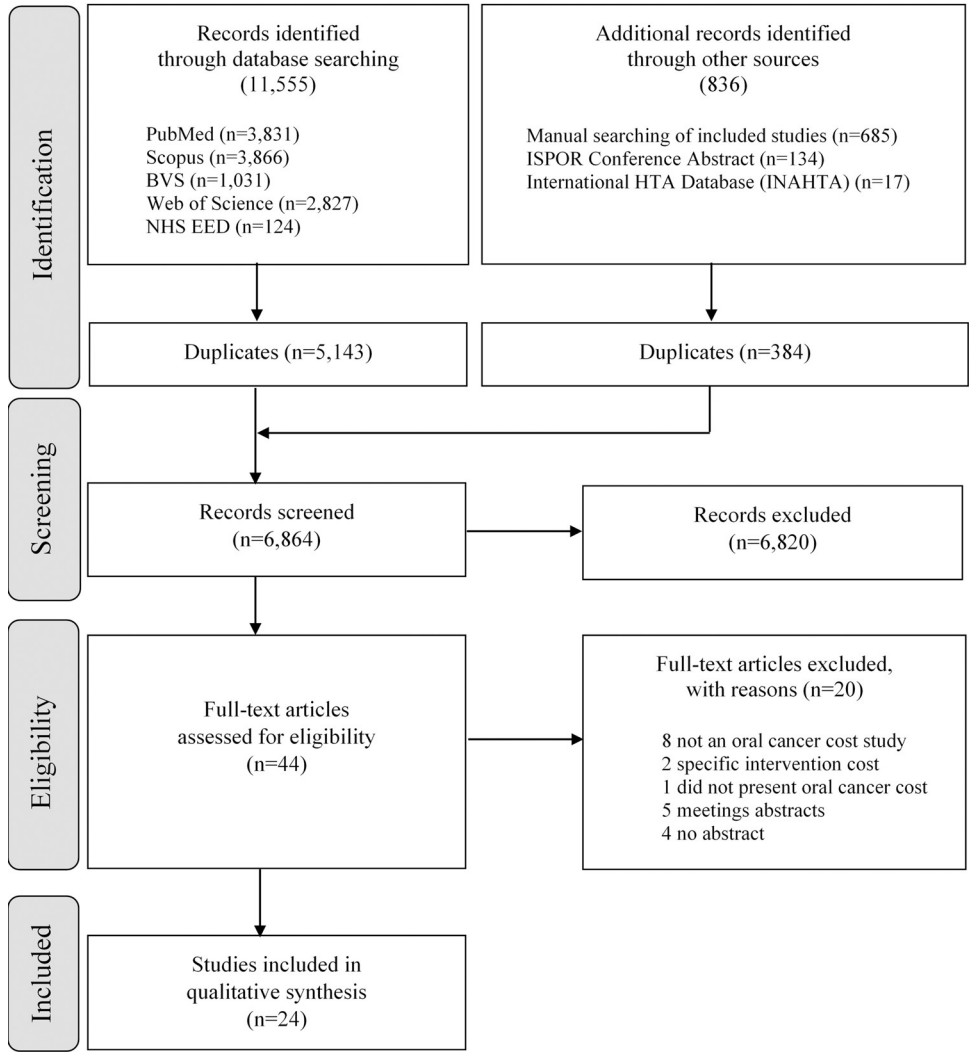

**Fig 1. Data acquisition flowchart.**

## Quality assessment

The global quality score of the studies, considered as the percentage rate of compliance to the items of the quality evaluation instrument, was 47.8% (SD = 10.9). The quality score varied from 38% [20, 32] to 66% [19] (Table 3). Regarding the study designs, the average of quality scores was 49.1% (SD = 9.9) for longitudinal studies, 47.3% (SD = 5.8) for cross-sectional and case control studies, and 46.0% (SD = 7.2) for studies based on information system data. No statistically significant difference was found among the average scores by study design (p = 0.796). Considering all studies, the Analytical Framework domain had an average score of 68.8% (SD = 15.0), the Methodology and Data domain 42.9% (SD = 10.1), and the Analysis and Reporting domains 43.8% (SD = 16.1), presenting a statistically significant difference among the average scores (p<0.001). The average of the quality scores of cross-sectional studies and those studies based on system data differed among domains (respectively p<0.001 and p = 0.001). The Analytical Framework domain had the highest average score in each included publication, when compared to the other two domains.

**Table 1. Summary of main characteristics of oral cancer cost studies from 2001 to 2020 (n = 24).**

| Study | Country | Sample (size, age, sex) | Cancer anatomical sites |
|---|---|---|---|
| **Longitudinal studies** | | | |
| 1. Kim, 2011 [23] | UK | 11,403 (mean age 63.2 years old; female (30.2%); male (69.8%)) | Lip; other and unspecified parts of tongue; oral cavity; pharynx; larynx |
| 2. Polesel, 2019 [26] | Italy | 879 (18–54 (20.6%); 55–59 (18.8%); 60–64 (19.1%) 65–69 (22.4%); 70–75 (19.1%)); female (19.2%); male (80.8%) | Oral cavity; oropharynx; hypopharynx; larynx |
| 3. Jacobson, 2012 [24] | USA | 6,812 (mean age: commercial 53.42; medicare 74.51; medicaid 53.36 years old; male: commercial (68.7%); medicare (65.4%); medicaid (58.8%) | Lip; base of tongue; gum; floor of mouth; other and unspecified parts of mouth; oral cavity; salivary gland cancer; major salivary gland; oropharynx |
| 4. Pollaers, 2019 [25] | Australia | 113 (mean age 60 years old; female (39.0%); male (61.0%)) | Other and unspecified parts of tongue; floor of mouth; retromolar trigone |
| 5. Huang, 2020 [27] | Taiwan | 50,784 (mean age 55.0 years old; female (9.0%); male (91.0%)) | Lip; base of tongue; other and unspecified parts of tongue; gum; floor of mouth; palate; other and unspecified parts of mouth; tonsil; oropharynx; pyriform sinus; hypopharynx; other and ill-defined sites in the lip, oral cavity, and pharynx is a medical classification as listed by WHO under the range |
| **Cross-sectional and case control studies** | | | |
| 6. Rezapour, 2018 [29] | Iran | 3,024 (mean age 55.27 years old; female (37.8%); male (62.2%)) | Lip; other and unspecified parts of tongue; floor of mouth; buccal |
| 7. van Aghtoven, 2001 [35] | The Netherlands | 854 (not informed) | Oral cavity; oropharynx; larynx |
| 8. Fisher, 2018 [36] | USA | 462 (mean age 61.1 years old; female (19.3%); male (80.7%)) | Oral cavity; oropharynx; hypopharynx/larynx; salivary glands; nasopharynx; other/unknown |
| 9. Nijdam, 2005 [32] | The Netherlands | 344 (mean age 56 (34–87) years-old; male (62.5%) and female (37.5%)) | Oropharynx |
| 10. Amarasinghe, 2019 [18] | Sri Lanka | 69 (40–50 (24.6%); 50–60 (43.5%); 60–70 (17.4%); >70 (14.5%)) | Lip; other and unspecified parts of tongue; floor of mouth; palate; other and unspecified parts of mouth |
| 11. Goyal, 2014 [33] | India | 100 (mean age 50.17 years old; female (8.0%); male (92.0%)) | Lip; other and unspecified parts of tongue; floor of mouth; buccal mucosa; retromolar trigone |
| 12. Zavras, 2002 [37] | Greece | 95 (not informed) | Upper lip, inner aspect; lower lip, inner aspect; lip, unspecified, inner aspect; commissure of lip; overlapping lesion of lip; lip, unspecified; base of tongue; other and unspecified parts of tongue; gum; floor of mouth; palate; other and unspecified parts of mouth |
| 13. van der Linden, 2016 [38] | The Netherlands | 125 (72% <65 years old, 28% > 65 years old) | Oral cavity; oropharynx; hypopharynx; nasopharynx; larynx |
| 14. Epstein, 2008 [34] | USA | 695 (mean age 63.49 years old; female (53.7%); male (46.3%)) | Lip; other and unspecified parts of tongue; gum; floor of mouth; other and unspecified parts of mouth; oropharynx; pharynx; nasopharynx; hypopharynx; Waldeyer's ring |
| 15. Lafuma, 2019 [30] | France | 267 (mean age 62 years old; female (15.0%); male (85.0%)) | Base of tongue; gum; floor of mouth; palate; tonsil; oropharynx; nasopharynx; piriform sinus; hypopharynx; larynx |
| 16. Patterson, 2020 [31] | Global stratified by region and income | 195 countries (data repositories) (not informed) | Lip; palate; base of tongue; other and unspecified parts of tongue; gum; floor of mouth; oral cavity; other and unspecified parts of mouth; tonsil; oropharynx; other pharynx; nasopharynx; larynx; thyroid |
| 17. Han, 2010 [39] | China | 456 (mean age 54.63 years old; female (38.6%); male (61.4%)) | Lip; other and unspecified parts of tongue; floor of mouth; palate; oral cavity; buccal mucosa, gingival tissues; retromolar trigon |
| 18. Enomoto, 2015 [40] | USA | 7,383 (female (38.8%); male (61.2%)) | Lip; base of tongue; other and unspecified parts of tongue; gum; floor of mouth; tonsil; oropharynx; salivary gland; nasopharynx; hypopharynx |
| 19. Lairson, 2017 [41] | USA | 934 (mean age 54 years old; female (18.2%); male (81.8%)) | Base of tongue; soft palate; uvula; lingual tonsil; oropharynx; pharynx otherwise unspecific |

*(Continued)*

**Table 1.** (Continued)

| Study | Country | Sample (size, age, sex) | Cancer anatomical sites |
|---|---|---|---|
| 20. Kim, 2020 [19] | Korea | 62,265 (not informed) | Lip; base of tongue; other and unspecified parts of tongue; gum; floor of mouth; palate; other and unspecified parts of mouth; tonsil; oropharynx; pyriform sinus; hypopharynx; other and ill-defined sites in the lip, oral cavity and pharynx is a medical classification as listed by WHO under the range |
| Cross-sectional studies based on information system data | | | |
| 21. Vatanasapt, 2012 [20] | Thailand | 207,439 visits (outpatient) and 8,360 admissions (inpatient) (not informed) | Other and unspecified parts of mouth; oral cavity; oropharynx; nasopharynx; hypopharynx; pharynx; larynx; parathyroid gland; external and middle ear; malignant melanoma; non-melanoma skin cancer; benign neoplasms |
| 22. Klussmann, 2013 [28] | Germany | 63,857 hospitalizations, 4,898 inpatient rehabilitations, and 17,494 sick leaves (age group: 15–80 years old; female (20.0%); male (80.0%)) | Base of tongue; other and unspecified parts of tongue; gum; floor of mouth; palate; other and unspecified parts of mouth; oral cavity; tonsil; oropharynx; pharynx; nasopharynx; larynx |
| 23. Keeping, 2018 [21] | England | 21,498 attendances (outpatient) and 27,326 hospital spells (inpatient) per year (not informed) | Lip; base of tongue; other and unspecified parts of tongue; gum; floor of mouth; palate; oral cavity; other and unspecified parts of mouth; tonsil; oropharynx; larynx |
| 24. Milani, 2021 [22] | Brazil | 117,317 admissions and 6,22,236 outpatient procedures (not informed) | Lip; base of tongue; other and unspecified parts of tongue; gum; floor of mouth; palate; other and unspecified parts of mouth; tonsil; oropharynx |

## Cost components

Fourteen studies [18, 19, 21–23, 25–27, 32, 34, 35, 37, 39, 41] evaluated all components of direct medical costs (surgery, chemotherapy, radiotherapy, follow-up, medications, exams), and only six [18, 19, 22, 29, 30, 39] investigated non-medical costs. Regarding indirect costs, three studies [18, 28, 30] evaluated absenteeism costs, two evaluated both [19, 29] absenteeism and early death, and one investigated early death costs [31]. None of them estimated presenteeism costs (Table 4).

Studies that met the inclusion criteria presented the estimates of oral cancer cost according to a wide variety of aspects: cost per patient, monthly cost, total cost in a period, cost per treatment or procedure, from the payer's perspective, cost components, outpatient and inpatient cost, services, by International Classification of Diseases, Tenth Revision (ICD-10) separately or in aggregate, by disease stage, follow-up, and disease recurrence (Table 5).

The OC cost comparison among studies was not possible. The set of anatomical sites investigated varied widely, in addition to the different measurement and costing methods. Only one of the studies included presented costs, separately, related to the sites considered as oral cancer in this systematic review (lip, oral cavity, and oropharynx) [22]. Most of the studies investigated lip, oral cavity, and oropharynx cancers together with other types of cancers from head and neck region (Table 5).

Only two studies [22, 23] investigated the cost of lip cancer separately from other ICD-10. The cost of lip cancer was estimated at GBP5,790 pounds per patient, over a five-year follow-up in the UK. In Brazil, the total expenditure of lip cancer was I$22.7 million in the period of nine years, from which I$18.1 million for inpatient costs and I$4.6 million for outpatient costs [22] (Table 5).

Three studies [23, 26, 35] showed the cost of oral cavity cancer per patient, estimated at GBP25,311 pounds in five years of follow-up in the UK [23]; EUR18,462 in two years of follow-up [26] in Italy; and EUR35,541 in a mathematical model estimated for 10 years in the Netherlands [35]. Two studies [20, 30] estimated oral cavity cancer cost per hospitalization, by information system data, with an average of THB29,531 [20] for Thailand and EUR6,482 for Germany [28], both over a one-year follow-up. One Brazilian study presented the expenditure

**Table 2. Methodological characteristics of oral cancer cost studies from 2001 to 2020.**

| Study | Type of study* | Study design | Data source | Cost-of-illness based approach | Estimation of resources and cost | Time horizon (years) | Perspective |
|---|---|---|---|---|---|---|---|
| Longitudinal studies | | | | | | | |
| 1. Kim, 2011 [19] | Cost-of-illness | Retrospective longitudinal | Secondary data | Incidence | Single study-based | 5 | Government |
| 2. Polesel, 2019 [26] | Cost-of-illness | Prospective longitudinal | Secondary data | Incidence | Single study-based | 2 | Unclear |
| 3. Jacobson, 2012 [24] | Cost analysis | Retrospective longitudinal | Secondary data | Incidence | Single study-based | 1 | Government and health insurance provider |
| 4. Pollaers, 2019 [25] | Cost analysis | Retrospective longitudinal | Secondary data | Incidence | Single study-based | 5 | Government |
| 5. Huang, 2020 [27] | Cost-of-illness | Retrospective longitudinal | Secondary data | Incidence | Single study-based | 5 | Societal and Government |
| Cross-sectional and case control studies | | | | | | | |
| 6. Rezapour, 2018 [29] | Cost-of-illness | Cross-sectional | Primary and secondary data | Prevalence | Single study-based | 1 | Societal |
| 7. van Aghtoven, 2001 [35] | Cost analysis | Cross-sectional | Primary and secondary data | Prevalence | Model-based | 10 | Hospital |
| 8. Fisher, 2018 [36] | Cost analysis | Cross-sectional | Primary and secondary data | Prevalence | Single study-based | 3 | Government and health insurance provider |
| 9. Nijdam, 2005 [32] | Cost-of-illness | Cross-sectional | Primary data | Prevalence | Single study-based | 4 | Hospital |
| 10. Amarasinghe, 2019 [18] | Cost-of-illness | Cross-sectional | Primary data | Prevalence | Single study-based | 1 | Societal |
| 11. Goyal, 2014 [33] | Cost analysis | Cross-sectional | Primary data | Prevalence | Single study-based | ~1 year 9 months | Hospital |
| 12. Zavras, 2002 [37] | Cost analysis | Cross-sectional | Primary data | Not clear | Single study-based | at least 6 months | Hospital |
| 13. van der Linden, 2016 [38] | Cost analysis | Cross-sectional | Primary data | Not clear | Single study-based | 1 | Hospital |
| 14. Epstein, 2008 [34] | Cost analysis | Cross-sectional | Secondary data | Prevalence | Single study-based | 1 | Government |
| 15. Lafuma, 2019 [30] | Cost-of-illness | Cross-sectional | Secondary data | Prevalence | Single study-based | 5 | Societal |
| 16. Patterson, 2020 [31] | Cost-of-illness | Cross-sectional | Secondary data | Incidence | Model-based | 1 | Societal |
| 17. Han, 2010 [39] | Cost analysis | Cross-sectional | Secondary data | Prevalence | Single study-based | 1 | Hospital |
| 18. Enomoto, 2015 [40] | Cost analysis | Cross-sectional | Secondary data | Prevalence | Single study-based | 1 | Government |
| 19. Lairson, 2017 [41] | Cost-of-illness | Case-control | Secondary data | Not clear | Single study-based | 2 | Government and health insurance provider |
| 20. Kim, 2020 [19] | Cost-of-illness | Cross-sectional | Secondary data | Prevalence | Single study-based | 5 | Societal and Government |
| Cross-sectional studies based on information system data | | | | | | | |
| 21. Vatanasapt, 2012 [20] | Cost-of-illness | Cross-sectional | Secondary data | Incidence | Single study-based | 1 | Government |
| 22. Klussmann, 2013 [28] | Cost-of-illness | Cross-sectional | Secondary data | Prevalence | Single study-based | 1 | Societal |
| 23. Keeping, 2018 [21] | Cost-of-illness | Cross-sectional | Secondary data | Prevalence | Single study-based | 5 | Hospital |

*(Continued)*

**Table 2.** (Continued)

| Study | Type of study* | Study design | Data source | Cost-of-illness based approach | Estimation of resources and cost | Time horizon (years) | Perspective |
|---|---|---|---|---|---|---|---|
| 24. Milani, 2021 [22] | Cost-of-illness | Cross-sectional | Secondary data | Prevalence | Single study-based | 9 | Government |

* We classified the type of study according to the comprehensiveness of the cost estimation. If the cost estimation was restricted to a small sample, the study was classified as cost analysis, generally a group of patients from one hospital; and if the cost estimation included a city, state or country, the study was classified as cost-of-illness.

of oral cavity cancer, in 9 years, as I\$257.1 million: on average I\$139.1 million for inpatients and I\$118.0 million for outpatients [22] (Table 5).

Three studies [26, 35, 41] showed the cost of oropharynx cancer per patient, estimated at EUR24,253 euros after two years of follow-up in Italy [26], EUR35,642 in a probabilistic mathematical model estimated for 10 years [35] in the Netherlands, which presented the health state after year 2, after year 4 and after years 5–10, calculated from the date of the primary diagnosis, and USD134,454 over a period of two years in the USA [41]. Three studies [20, 22, 28] estimated oropharynx cancer per hospitalization, by information system data, with an average of THB26,331 [20] in Thailand and EUR4,268 in Germany [28], both over a one-year follow-up, and I\$1,338 in Brazil over nine years [22] (Table 5).

Only four studies showed the costs of OC by cost components (direct and indirect costs) [28–30]. In France (2018) [30], the direct medical cost of head and neck cancer was EUR49,954 per patient, considering outpatient and inpatient care, public hospitals services, and private for-profit hospitals services. The indirect cost was EUR2,989 per patient for disability and sick leave. In Germany (2008) [28], the direct medical cost of oral cancer was approximately EUR113 million, and the indirect cost was EUR18 million (sick leave). The direct medical cost of oropharyngeal cancer was approximately EUR83 million, and the indirect cost was EUR16 million (sick leave) [28]. In Iran [29], the cost of lip cancer, and for other and unspecified parts of tongue, the floor of the mouth, and buccal cancers was approximately USD27 million, USD5 million, and USD32 million for direct and direct non-medical and indirect costs, respectively [29]. The direct medical costs in Taiwan (2018) [27] were USD19,644 per patient and indirect costs for morbidity and mortality were USD1,286 and USD35,570 per patient, respectively, in a follow-up over 2.3 years (Table 5).

The LC burden of cost was 18.3% of UK GDP per capita [23]. Regarding the OCC cost, the burden was 79.8%, 64.9%, and 79.8% of UK, Italian, and the Netherlands' GDP per capita, respectively [23, 26, 35]. The OPC burden of cost was 85.2%, 80.3%, 215.0% of Italy, the Netherlands, and the USA GDP per capita, respectively [26, 35, 41] (Table 6).

Five studies showed outpatient and inpatient costs [22, 23, 25, 30, 41]. In general, inpatient costs are higher than outpatient costs, with a coefficient of variation of 93% [30] to 967.5% [23]. Outpatient costs exceeded inpatient costs in those studies in which chemotherapy and radiotherapy procedures were performed as outpatient costs [22, 41] (Fig 2A).

Regarding the resource quantification, most of the included studies used a top-down approach (18 studies), generally obtained by allocating portions of a known total expenditure to a specific disease stratified by type of cost. Only 6 studies relied on individual data (bottom-up approach), generally obtained by multiplying the unit costs by quantities.

Advanced staging was more expensive (from 21.9% to 373.3%) than early cancer staging [18, 25, 29, 34, 37], despite the lack of a clinical stage standard definition of the disease and the different sets of head and neck tumors studied (Fig 2B).

**Table 3. Assessment of quality of studies of oral cancer cost included in the systematic review, according to the critical guide of Larg & Moss, 2011 [14].**

| Domains | Item | Longitudinal studies | | | | | Cross-sectional and case control studies | | | | | | | | | | | | | | | Cross-sectional studies based on information system data | | | |
|---|---|---|---|---|---|---|---|---|---|---|---|---|---|---|---|---|---|---|---|---|---|---|---|---|---|
| | | Kim (2011) | Polesel (2019) | Jacobson (2012) | Pollaers (2019) | Huang (2020) | Rezapour (2018) | van Agthoven (2001) | Fisher (2018) | Nijdam (2005) | Amarasinghe (2019) | Goyal (2014) | Zavras (2002) | van der Linden (2016) | Epstein (2008) | Lafuma (2019) | Patterson (2020) | Han (2010) | Enomoto (2015) | Lairson (2017) | Kim (2020) | Vatanasapt (2012) | Klussmann (2013) | Keeping (2018) | Milani (2021) |
| **Analytical framework** | 1A Cost—perspective | ✓ | | ✓ | ✓ | ✓ | ✓ | ✓ | | ✓ | ✓ | ✓ | ✓ | | ✓ | ✓ | | ✓ | ✓ | ✓ | | | ✓ | ✓ | ✓ |
| | 1B Cost—epidemiological approach | ✓ | ✓ | ✓ | ✓ | | ✓ | ✓ | ✓ | ✓ | ✓ | ✓ | ✓ | ✓ | ✓ | ✓ | ✓ | ✓ | ✓ | ✓ | ✓ | | ✓ | ✓ | ✓ |
| | 1CI Cost—societal | | | | ✓ | | ✓ | | | | ✓ | ✓ | | ✓ | ✓ | | | ✓ | | | ✓ | ✓ | ✓ | ✓ | ✓ |
| | 1CII Cost—timeframe | ✓ | | ✓ | | ✓ | ✓ | ✓ | ✓ | ✓ | ✓ | ✓ | | ✓ | | ✓ | | ✓ | ✓ | | ✓ | ✓ | ✓ | ✓ | ✓ |
| | 1CIII Cost—risk factor | NA | NA | NA | NA | NA | NA | NA | NA | NA | NA | NA | NA | NA | NA | NA | NA | NA | NA | NA | NA | NA | NA | NA | NA |
| | 1CIV Cost—counterfactual population | | ✓ | ✓ | | | | | | | | | | | ✓ | | | | ✓ | ✓ | | | | | |
| | **Total number of positive and NA answers of the Analytical frame domain (%)** | 4 (67) | 3 (50) | 5 (83) | 4 (67) | 3 (50) | 5 (83) | 4 (67) | 3 (50) | 4 (67) | 5 (83) | 5 (83) | 3 (50) | 4 (67) | 5 (83) | 4 (67) | 2 (33) | 5 (83) | 5 (83) | 4 (67) | 4 (67) | 3 (50) | 5 (83) | 5 (83) | 5 (83) |
| | **Positive and NA answers of the of the Analytical frame domain (average ± SD) (68.8±15.0) ** p = 0.529** | 63.3 ± 13.9 | | | | | 68.9 ± 15.3 | | | | | | | | | | | | | | | 75.0 ± 16.7 | | | |
| **Methodology and data** | 2AI Quantification—additional cost | ✓ | ✓ | ✓ | | ✓ | ✓ | ✓ | | ✓ | | ✓ | ✓ | ✓ | ✓ | ✓ | ✓ | ✓ | ✓ | ✓ | ✓ | | | ✓ | ✓ |
| | 2AII Quantification—confounders controlled | ✓ | ✓ | | ✓ | ✓ | ✓ | ✓ | ✓ | ✓ | ✓ | ✓ | ✓ | ✓ | ✓ | ✓ | ✓ | ✓ | ✓ | ✓ | ✓ | ✓ | ✓ | ✓ | ✓ |
| | 2AIII Quantification—important effects | | ✓ | | ✓ | ✓ | ✓ | ✓ | ✓ | ✓ | ✓ | ✓ | ✓ | ✓ | ✓ | | | ✓ | | | | ✓ | ✓ | ✓ | ✓ |
| | 2AIV Quantification—differences across subpopulations | | | | ✓ | ✓ | ✓ | ✓ | ✓ | | ✓ | ✓ | ✓ | | ✓ | | | ✓ | | | | | | | ✓ |
| | 2AV Quantification—required level of detail | ✓ | ✓ | ✓ | ✓ | ✓ | ✓ | ✓ | ✓ | | ✓ | ✓ | ✓ | ✓ | ✓ | ✓ | ✓ | ✓ | ✓ | ✓ | ✓ | ✓ | ✓ | ✓ | ✓ |
| | 2BI Resource quantification—population based (top-down) | ✓ | ✓ | ✓ | ✓ | ✓ | ✓ | ✓ | ✓ | ✓ | ✓ | ✓ | ✓ | | ✓ | ✓ | ✓ | ✓ | ✓ | ✓ | | ✓ | ✓ | ✓ | |
| | 2BII Resource quantification—person based (bottom-up) | | | | | | | | ✓ | | ✓ | | | | | | | | | | | | | ✓ | |
| | 2BIII Resource quantification—data representative | | | | | | | | | | | | | | | | | | | | ✓ | | | | |
| | 2BIV Resource quantification—other relevant issue (model-based) | | | | ✓ | | | | | | | | | | | | ✓ | | | | | | | | |
| | 2C Resource—healthcare | | | | ✓ | | ✓ | | ✓ | | ✓ | | | | | ✓ | | | | | ✓ | | | | ✓ |
| | 2D Productivity—losses and assumptions | | | | | ✓ | | | | | ✓ | | | | | ✓ | ✓ | | | | ✓ | | ✓ | | ✓ |
| | 2EI Intangible costs—mortality-related losses avoided | | | | | | | | | | | | | | | | | | | | | | | | |
| | 2EII Intangible costs—study's perspective losses | | | | | | | | | | | | | | | | | | | | | | | | |
| | **Total number of positive and NA answers of the Methodology and data domain (%)** | 4 (31) | 5 (38) | 3 (23) | 7 (54) | 7 (54) | 7 (54) | 6 (46) | 7 (54) | 4 (31) | 8 (62) | 6 (46) | 6 (46) | 4 (31) | 6 (46) | 6 (46) | 6 (46) | 6 (46) | 4 (31) | 4 (31) | 6 (46) | 4 (31) | 5 (38) | 6 (46) | 7 (54) |
| | **Positive and NA answers of the of the Methodology and data domain (average ± SD) (42.9±10.1) ** p = 0.745** | 40.0 ± 13.8 | | | | | 44.1 ± 9.4 | | | | | | | | | | | | | | | 42.3 ± 9.9 | | | |

*(Continued)*

**Table 3.** (Continued)

| Domains | Item | | Longitudinal studies | | | | | Cross-sectional and case control studies | | | | | | | | | | | | | | | Cross-sectional studies based on information system data | | | |
|---|---|---|---|---|---|---|---|---|---|---|---|---|---|---|---|---|---|---|---|---|---|---|---|---|---|---|---|
| | | | Kim (2011) | Polesel (2019) | Jacobson (2012) | Pollaers (2019) | Huang (2020) | Rezapour (2018) | van Aghtoven (2001) | Fisher (2018) | Nijdam (2005) | Amarasinghe (2019) | Goyal (2014) | Zavras (2002) | van der Linden (2016) | Epstein (2008) | Lafuma (2019) | Patterson (2020) | Han (2010) | Enomoto (2015) | Lairson (2017) | Kim (2020) | Vatanasapt (2012) | Klussmann (2013) | Keeping (2018) | Milani (2021) |
| Analysis and reporting | 3A | Study question answered | ✓ | ✓ | ✓ | ✓ | ✓ | ✓ | ✓ | ✓ | ✓ | ✓ | ✓ | ✓ | ✓ | ✓ | ✓ | ✓ | ✓ | ✓ | ✓ | ✓ | ✓ | ✓ | ✓ | ✓ |
| | 3B | Range of estimates presented | | ✓ | | | | | | | | | | | | | | | | | | | | | | |
| | 3C | No main uncertainties identified | | ✓ | | | | | | | | | | | | | | | | | | ✓ | | | | |
| | 3DI | Sensitivity—analysis performed | | ✓ | | | | | | | | | | | | | | | | | | ✓ | | | | |
| | 3DII | Sensitivity—key assumptions | | ✓ | | | | | | | | | | | | | | | | | | ✓ | | | | |
| | 3DIII | Sensitivity—point estimates | | ✓ | | | | | | | | | | | | | | | | | | ✓ | | | | |
| | 3E | Adequate documentation—cost components, data, sources, assumptions, and methods | ✓ | ✓ | ✓ | ✓ | ✓ | ✓ | ✓ | ✓ | ✓ | ✓ | ✓ | ✓ | ✓ | ✓ | ✓ | ✓ | ✓ | ✓ | ✓ | ✓ | ✓ | ✓ | ✓ | ✓ |
| | 3F | Uncertainty—estimates discussed | | ✓ | | | | | | | | | | | | | | | | | | ✓ | | | | |
| | 3G | Limitations | ✓ | ✓ | ✓ | ✓ | ✓ | ✓ | ✓ | ✓ | | ✓ | | ✓ | ✓ | ✓ | ✓ | ✓ | ✓ | ✓ | ✓ | ✓ | ✓ | ✓ | ✓ | ✓ |
| | 3H | Results—appropriate level of detail | ✓ | ✓ | ✓ | ✓ | ✓ | ✓ | ✓ | ✓ | ✓ | ✓ | ✓ | ✓ | ✓ | ✓ | ✓ | ✓ | ✓ | ✓ | ✓ | ✓ | ✓ | ✓ | ✓ | ✓ |
| | | Total number of positive and NA answers of the Analysis and reporting domain (%) | 4 (40) | 10 (100) | 4 (40) | 4 (40) | 4 (40) | 4 (40) | 4 (40) | 4 (40) | 3 (30) | 4 (40) | 3 (30) | 4 (40) | 4 (40) | 4 (40) | 4 (40) | 4 (40) | 4 (40) | 4 (40) | 4 (40) | 9 (90) | 4 (40) | 4 (40) | 4 (40) | 4 (40) |
| | | Positive and NA answers of the the Analysis and reporting domain (average ± SD) (43.8±16.1) ** p = 0.443 | | 52.0 ± 26.8 | | | | | | | | | | | 42.0 ± 13.7 | | | | | | | | | 40.0 ± 0.0 | | |
| | | Total number of positive and NA answers (%) | 12 (41) | 18 (62) | 12 (41) | 15 (52) | 14 (48) | 16 (55) | 14 (48) | 14 (48) | 11 (38) | 17 (59) | 13 (45) | 12 (41) | 12 (41) | 15 (52) | 14 (48) | 12 (41) | 16 (55) | 13 (45) | 12 (41) | 19 (66) | 11 (38) | 14 (48) | 15 (52) | 16 (55) |
| | | **Global score: 47.8% ± 10.9** | | | | | | | | | | | | | | | | | | | | | | | | |
| | | Positive and NA answers per type of study groups (average ± SD) ** p = 0.796 | | 49.1 ± 9.9 | | | | | | | | | | | 47.3 ± 5.8 | | | | | | | | | 46.0 ± 7.2 | | |
| | | ANOVA test per domain (p-value) | | 0.207 | | | | | | | | | | | <0.001 | | | | | | | | | 0.001 | | |

SD: standard deviation.

** ANOVA test (p<0.05).

**Table 4. Estimates of medical, non-medical, and indirect costs of oral cancer presented in the methods of studies from 2001 to 2020.**

| Study | Direct costs | | | | | | | | Indirect cost | |
|---|---|---|---|---|---|---|---|---|---|---|
| | Medical costs | | | | | | Non-medical costs | Absenteeism | Early Death |
| | Surgery | Chemotherapy | Radiotherapy | Follow up | Medications | Exams | | | |
| Longitudinal studies | | | | | | | | | | |
| 1. Kim, 2011 [23] | ✓ | ✓ | ✓ | ✓ | ✓ | ✓ | | | | |
| 2. Polesel, 2019 [26] | ✓ | ✓ | ✓ | ✓ | ✓ | ✓ | | | | |
| 3. Jacobson, 2012 [24] | ✓ | ✓ | ✓ | ✓ | ✓ | | | | | |
| 4. Pollaers, 2019 [25] | ✓ | ✓ | ✓ | ✓ | ✓ | ✓ | | | | |
| 5. Huang, 2020 [27] | ✓ | ✓ | ✓ | ✓ | ✓ | ✓ | | | | |
| Cross-sectional and case control studies | | | | | | | | | | |
| 6. Rezapour, 2018 [29] | ✓ | ✓ | ✓ | ✓ | | ✓ | ✓ | ✓ | ✓ |
| 7. van Aghtoven, 2001 [35] | ✓ | ✓ | ✓ | ✓ | ✓ | ✓ | | | |
| 8. Fisher, 2018 [36] | | ✓ | | ✓ | ✓ | | | | |
| 9. Nijdam, 2005 [32] | ✓ | ✓ | ✓ | ✓ | ✓ | ✓ | | | |
| 10. Amarasinghe, 2019 [18] | ✓ | ✓ | ✓ | ✓ | ✓ | ✓ | ✓ | ✓ | |
| 11. Goyal, 2014 [33] | ✓ | ✓ | ✓ | ✓ | | | | | |
| 12. Zavras, 2002 [37] | ✓ | ✓ | ✓ | ✓ | ✓ | ✓ | | | |
| 13. van der Linden, 2016 [38] | ✓ | ✓ | ✓ | | ✓ | ✓ | | | |
| 14. Epstein, 2008 [34] | ✓ | ✓ | ✓ | ✓ | ✓ | | | | |
| 15. Lafuma, 2019 [30] | | ✓ | | ✓ | ✓ | ✓ | ✓ | ✓ | |
| 16. Patterson, 2020 [31] | | | | | | | | | ✓ |
| 17. Han, 2010 [39] | ✓ | ✓ | ✓ | ✓ | ✓ | ✓ | ✓ | | |
| 18. Enomoto, 2015 [40] | | | | ✓ | | | | | |
| 19. Lairson, 2017 [41] | ✓ | ✓ | ✓ | ✓ | ✓ | ✓ | | | |
| 20. Kim, 2020 [19] | ✓ | ✓ | ✓ | ✓ | ✓ | ✓ | ✓ | ✓ | ✓ |
| Cross-sectional studies based on information system data | | | | | | | | | | |
| 21. Vatanasapt, 2012 [20] | ✓ | | | ✓ | | ✓ | | | |
| 22. Klussmann, 2013 [28] | ✓ | ✓ | ✓ | | ✓ | ✓ | | ✓ | |
| 23. Keeping, 2018 [21] | ✓ | ✓ | ✓ | ✓ | ✓ | ✓ | | | |
| 24. Milani, 2021 [22] | ✓ | ✓ | ✓ | ✓ | ✓ | ✓ | ✓ | | |

The treatment of recurrent squamous cell carcinoma of the floor of mouth, tongue, and alveolar trigone was 51% more expensive than the treatment of primary tumors, in a two-year follow-up study [25].

## Discussion

Our systematic review highlights the economic impact of oral cancer as a rising burden from a worldwide perspective. In a resource-scarce healthcare environment, with an aging population and an increasing number of new diagnoses of oral cancer, this new knowledge is imperative in guiding resource allocation for oral cancer care provision and research funding. Deployment of interventions to improve outcomes for patients should be measured not only in terms of clinical outcome, but also in terms of economic impact. Furthermore, the analysis uncovers the large heterogeneity of cost of illness studies (COI) focused on oral cancer.

In some western countries, the economic burden of OCC and OPC is more than 60% of GDP per capita [23, 26, 35], reaching 215% of US GDP per capita (OPC) [41]. Considering that the GDP per capita corresponds to the average income of families [16], it is a cost that the individual cannot, in most cases, bear alone, and which requires the support of governments.

**Table 5. Summary of costs estimates of included studies in the systematic review.**

| Study | Year of cost | Country | Currency | Cancer anatomical sites considered for cost | Cost per patient (95% CI) | Cost per patient by clinical stage | Other measures of cost (95% CI) |
|---|---|---|---|---|---|---|---|
| | | | | Longitudinal studies | | | |
| 1. Kim, 2011 [23] | 2009 | United Kingdom | Pounds (GBP) | Lip; other and unspecified parts of tongue; oral cavity; pharynx; larynx | Post-operative treatment for resected patients (average)– 5-year follow-up:<br>• Lip: 5,790<br>• Tongue: 19,493<br>• Oral cavity: 25,311 | Not applicable | Post-operative treatment for resected patients (average)– 5-year follow-up:<br>Inpatient care:<br>• Lip: 4,798<br>• Tongue: 17,910<br>• Oral cavity: 23,143<br>Outpatient care:<br>• Lip: 992<br>• Tongue: 1,583<br>• Oral cavity: 2,168 |
| 2. Polesel, 2019 [26] | 2010 | Italy | Euros (EUR) | Oral cavity (including lip and pharynx); oropharynx; hypopharynx; larynx | Cost per patient (average– 95% CI)– 2-year follow-up:<br>• Oral cavity: 18,462 (17,720–19,205)<br>• Oropharynx: 24,253 (23,197–25,310) | Not applicable | Cost per patient (average– 95%CI):<br>Oral cavity:<br>• Before treatment: 1,223 (1,103–1,343)<br>• 0–3 months treatment: 11,102 (10,702–11,503)<br>• 4–12 months treatment: 4,421 (3,838–5,004)<br>• 13–24 months treatment: 2,282 (2,071–2,493)<br>Oropharynx:<br>• Before treatment: 2,932 (2,640–3,223)<br>• 0–3 months treatment: 12,646 (12,123–13,170)<br>• 4–12 months treatment: 8,658 (7,562–9,754)<br>• 13–24 months treatment: 4,113 (3,696–4,531) |
| 3. Jacobson, 2012 [24] | 2009 | United States of America | US Dollars (USD) | Lip; base of tongue; gum; floor of mouth; other and unspecified parts of mouth; oral cavity; salivary gland; major salivary gland; oropharynx | Not reported | Not applicable | Oral cavity, oropharynx and salivary gland tumors (average ± standard deviation)– 1-year follow-up:<br>• Commercial insurance: 79,151 ± 86,170<br>• Medicare: 48,410 ± 61,599<br>• Medicaid: 59,404 ± 74,919 |
| 4. Pollaers, 2019 [25] | 2016/ 2017 | Australia | Australian Dollars (AUD) | Other and unspecified parts of tongue; floor of mouth; retromolar trigone | Squamous cell carcinoma (average)– 2-year follow-up:<br>Floor of mouth: 103,832<br>• First year: 101,187<br>• Second year: 2,645<br>Tongue: 92,761<br>• First year: 86,391<br>• Second year: 6,279 | Inpatients costs:<br>Stage I:<br>• 1-year: 33,985<br>• 2-year: 37,101<br>• 5-year: 43,661<br>Stage II:<br>• 1-year: 61,690<br>• 2-year: 45,376<br>• 5-year: 44,548<br>Stage III:<br>• 1-year: 79,684<br>• 2-year: 90,557<br>• 5-year: 88,976<br>Stage IVa:<br>• 1-year: 93,269<br>• 2-year: 104,257<br>• 5-year: 118,913 | Squamous cell carcinoma of floor of the mouth, tongue, and alveolar trigone (average)– 2-year follow-up:<br>• Remained disease-free: 65,012<br>• Cases with disease recurrence: 98,359<br>First year:<br>• Inpatient: 66,004<br>• Outpatient: 30,214<br>Second year:<br>• Inpatient: 72,208<br>• Outpatient: 35,590 |

*(Continued)*

**Table 5.** (Continued)

| Study | Year of cost | Country | Currency | Cancer anatomical sites considered for cost | Cost per patient (95% CI) | Cost per patient by clinical stage | Other measures of cost (95% CI) |
|---|---|---|---|---|---|---|---|
| **Longitudinal studies** | | | | | | | |
| 5. Huang, 2020 [27] | 2018 | Taiwan | US Dollars (USD) | Lip; base of tongue; other and unspecified parts of tongue; gum; floor of mouth; palate; other and unspecified parts of mouth; tonsil; oropharynx; piriform sinus; hypopharynx; ill-defined and unspecified sites of lips, oral cavity, and pharynx | Average per patient– 2.3-years follow-up: 56,501 Mean ± standard deviation: • Direct medical costs mean: 19,644 ± 15,305 • Indirect costs morbidity: 1,286 ± 1,386 • Indirect costs mortality: 35,570 ± 61,859 • All medication costs mean: 2,359 • Anticancer drug costs mean: 638 | --- | --- |
| **Cross-sectional and case control studies** | | | | | | | |
| 6. Rezapour, 2018 [29] | 2014 | Iran | US Dollars (USD) | Lip; other and unspecified parts of tongue; floor of mouth; buccal | Floor of mouth, lip, tongue, buccal (average)– 1-year follow-up: Direct costs: • Diagnosis: 82 • Early stage (I/II): 2,225 • Advanced stage (III/IV): 10,532 • Recurrency: 1,485 • Follow-up: 291 Direct non-medical costs • Traveling: 1,035 • Home care: 665 Indirect costs: • Employed patients: 2,477 • Unemployed patients: 1,230 • Accompanied: 481 • Premature mortality: 106,257 | Floor of mouth, lip, tongue, buccal (average)– 1-year follow-up: • Stages I/II (surgery and radiotherapy): 2,225 • Stages III/IV (surgery, radiotherapy, and chemotherapy): 10,532 | Total cost (Iran): 64,245,173 Direct costs: 27,284,501 • Direct non-medical costs: 5,143,629 • Indirect costs: 31,817,043 |
| 7. van Agthoven, 2001 [35] | 1996 | The Netherlands | Euros (EUR) | Oral cavity; oropharynx; larynx | Average per patient– 10-year follow-up: • Oral cavity: 35,541 • Oropharynx: 35,642 | Not applicable | Oral cavity (per patient): • Primary tumor: 25,425 • Recurrent tumor: 25,543 Oropharynx (per patient): • Primary tumor: 25,679 • Recurrent tumor: 25,145 |
| 8. Fisher, 2018 [36] | 2016 | United States of America | US Dollars (USD) | Oral cavity; oropharynx; hypopharynx/larynx; salivary glands; nasopharynx; other/ unknown | Not reported | Not applicable | Monthly health care costs (average ± standard deviation): Total costs: 14,391 ± 19,510 • Hospitalization: 8,136 ± 16,880 • Emergency visits: 433 ± 1,259 • Office visits: 123 ± 109 • Systemic anticancer therapy: 2,875 ± 5,259 • Medical oncology procedures: 2,415 ± 6,962 • Infused supportive care drugs: 232 ± 479 All other drugs delivered: 178 ± 242 |

*(Continued)*

**Table 5.** (*Continued*)

| Study | Year of cost | Country | Currency | Cancer anatomical sites considered for cost | Cost per patient (95% CI) | Cost per patient by clinical stage | Other measures of cost (95% CI) |
|---|---|---|---|---|---|---|---|
| | | | | **Longitudinal studies** | | | |
| 9. Nijdam, 2005 [32] | 2001 | The Netherlands | Euros (EUD) | Oropharynx | Oropharynx (average): • BT group*: 18,001 • S group*: 28,130 • EBRT group*: 21,143 | Not applicable | Not available |
| 10. Amarasinghe, 2019 [18] | 2016 | Sri Lanka | US Dollars (USD) | Lip; other and unspecified parts of tongue; floor of mouth; palate; other and unspecified parts of mouth | Lip, tongue, and mouth (average)– 1-year follow-up: Stage II: System cost: • Recurrent costs: 381 • Capital cost: 13 Household: • Direct costs: 256 • Indirect costs: 263 Stage III and IV: System cost: • Recurrent costs: 2,011 • Capital cost: 13 Household: • Direct costs: 217 • Indirect costs: 263 | Lip, tongue, and mouth per patient (average)– 1-year follow-up: • Stage II: 912 • Stage III/IV: 2,507 | Not available |
| 11. Goyal, 2014 [33] | 2011/2012 | India | Rupees (INR) | Lip; other and unspecified parts of tongue; floor of mouth; buccal mucosa; retromolar trigone | Hospitalization (average ± standard deviation): 1,46,092 ± 37,325 | Hospitalization (average): • Stage I: 1,49,995 • Stage II: 1,41,621 • Stage III: 1,82,859 | Not available |
| 12. Zavras, 2002 [37] | 2001 | Greece | US Dollars (USD) | Upper lip, inner aspect; lower lip, inner aspect; lip, unspecified, inner aspect; commissure of lip; overlapping lesion of lip; lip, unspecified; base of tongue; other and unspecified parts of tongue; gum; floor of mouth; palate; other and unspecified parts of mouth | Average per patient: 7,450 | Average: • Stage I: 3,662 • Stage II: 5,867 • Stage III: 10,316 • Stage IV: 11,467 | Not available |
| 13. van der Linden, 2016 [38] | 2013 | The Netherlands | Euros (EUR) | Oral cavity; oropharynx; hypopharynx; nasopharynx; larynx | Not reported | Not applicable | Cost per treatment group (average ± standard deviation): • Cisplatin + 5-fluorouracil + cetuximab: 39,459 ± 21,149 • Other platinum-based combination therapy: 38,584 ± 26,065 • Methotrexate monotherapy: 10,075 ± 9,891 • Capecitabine monotherapy: 10,585 ± 14,544 • Other: 17,506 ± 16,634 |

(*Continued*)

**Table 5.** (Continued)

| Study | Year of cost | Country | Currency | Cancer anatomical sites considered for cost | Cost per patient (95% CI) | Cost per patient by clinical stage | Other measures of cost (95% CI) |
|---|---|---|---|---|---|---|---|
| | | | | **Longitudinal studies** | | | |
| 14. Epstein, 2008 [34] | 2002 | United States of America | US Dollars (USD) | Lip; other and unspecified parts of tongue; gum; floor of mouth; other and unspecified parts of mouth; oropharynx; pharynx; nasopharynx; hypopharynx; Waldeyer's ring | Oral and pharyngeal squamous cell carcinoma (median– 95%CI)– 1-year follow-up: 25,319 (21,825–27,665) | Oral and pharyngeal squamous cell carcinoma (median): • No treatment 9,763 (IR: 3,520–24,439) • Early-stage treatment: 22,658 (IR: 10,425–42,664) • Late-stage treatment: 27,665 (IR: 19,335–52,547) | Not available |
| 15. Lafuma, 2019 [30] | 2018 | France | Euros (EUR) | Base of tongue; gum; floor of mouth; palate; tonsil; oropharynx; nasopharynx; piriform sinus; hypopharynx; larynx | Not reported | Not applicable | Squamous cell carcinomas of the head and neck (average–95%CI): Total costs: 52,943 Direct costs: 49,954 • Ambulatory care: 17,047 (14,941–19,152) • Inpatient care 32,908 (29,525–36,290) • Public hospitals: 26,015 (22,716–29,314) • Private for-profit hospitals: 6,892 (4,809–8,976) Indirect costs: 2,989 • Disability: 1,397 (624–2,171) • Sick leave: 1,592 (888–2,297) |
| 16. Patterson, 2020 [31] | 2017 | Global, stratified by region and income | Purchasing Power Parity (PPP) | Lip; palate; base of tongue; other and unspecified parts of tongue; gum; floor of mouth; oral cavity; other and unspecified parts of mouth; tonsil; oropharynx; other pharynx; nasopharynx; larynx; thyroid | Not reported | Not applicable | Projected global cumulative loss of 535 billion US dollars in economic output due to head and neck cancer between 2018 and 2030. Southeast Asia, East Asia, and Oceania will suffer the greatest GDP losses at 180 billion US Dollars, and South Asia will lose 133 billion US Dollars |
| 17. Han, 2010 [39] | 2007 | China | Chinese Yuans (CNY) | Lip; other and unspecified parts of tongue; floor of mouth; palate; oral cavity; buccal mucosa; gingival tissues; retromolar trigon | Squamous cell carcinoma (average ± standard deviation): 27,890 ± 11,032 | Not applicable | Squamous cell carcinoma (average ± standard deviation): • Diagnosis: 3,465 ± 1,059 • Treatment: 19,995 ± 9,701 • Hospitalization: 4,429 ± 1,618 |
| 18. Enomoto, 2015 [40] | 2009 | United States of America | US Dollars (USD) | Lip; base of tongue; other and unspecified parts of tongue; gum; floor of mouth; tonsil; oropharynx; salivary gland; nasopharynx; hypopharynx | Not reported | Not applicable | Cost per patient ≤ 30 days before death (average): Oral cavity: • Hospice: 7,880 • Non hospice: 14,990 Oropharynx: • Hospice: 8,790 • Non hospice: 16,390 |

*(Continued)*

**Table 5.** (Continued)

| Study | Year of cost | Country | Currency | Cancer anatomical sites considered for cost | Cost per patient (95% CI) | Cost per patient by clinical stage | Other measures of cost (95% CI) |
|---|---|---|---|---|---|---|---|
| **Longitudinal studies** | | | | | | | |
| 19. Lairson, 2017 [41] | 2015 | United States of America | US Dollars (USD) | Base of tongue; soft palate; uvula; lingual tonsil; oropharynx; pharynx otherwise unspecified | Oropharynx: 134,454 ± 108,635 | Not applicable | Cost per patient (average ± standard deviation) - 2-year follow-up: Oropharynx: • Inpatient: 24,341 ± 48,972 • Outpatient: 106,604 ± 82,221 • Outpatient prescription drugs: 3,550 ± 5,183 • Surgery: 8,320 ± 15,111 • Radiotherapy: 50,362 ± 28,928 • Chemotherapy: 3,277 ± 2,822 |
| 20. Kim, 2020 [19] | 2015 | Korea | US Dollars (USD) | Lip; base of tongue; other and unspecified parts of tongue; gum; floor of mouth; palate; other and unspecified parts of mouth; tonsil; oropharynx; piriform sinus; hypopharynx; ill-defined and unspecified sites of lips, oral cavity, and pharynx | *Average per patient: 20,107 | --- | Total cost: 1,248 billion • Male: 1,057 billion • Female: 190,9 million |
| **Cross-sectional studies based on information system data** | | | | | | | |
| 21. Vatanasapt 2012 [20] | 2010 | Thailand | Baths (THB) | Other and unspecified parts of mouth; oral cavity; oropharynx; nasopharynx; hypopharynx; pharynx; larynx; parathyroid gland; eternal and middle ear; malignant melanoma; non-melanoma skin cancer; benign neoplasms | Average per hospitalization (95%CI): • Oral cavity: 29,531 (28,316–30,745) • Oropharynx: 26,331 (23,995–28,668) • Ill-defined sites in oral cavity and pharynx: 19,356 (14,621–24,090) | Not applicable | Total cost of hospitalization: • Oral cavity: 191,685,473 (n = 6,491) • Oropharynx: 37,258,803 (n = 1,415) • Ill-defined sites in oral cavity and pharynx: 3,135,667 (n = 162) |
| 22. Klussmann, 2013 [28] | 2008 | Germany | Euros (EUR) | Base of tongue; other and unspecified parts of tongue; gum; floor of mouth; palate; other and unspecified parts of mouth; oral cavity; tonsil; oropharynx; pharynx; nasopharynx; larynx | * Average per hospitalization: Oral cavity: Direct costs: • Hospitalization: 6,482 • Inpatient rehabilitation: 2,713 Indirect costs: • Sick leave: 3,669 Oropharynx: Direct costs: • Hospitalization: 4,• 268 Inpatient rehabilitation: 2,705 Indirect costs: • Sick leave: 3,263 | Not applicable | *Total cost: Oral cavity: 131,019,446 • Direct costs: 113,268,892 ○ Hospitalization: 109,503,702 ○ Inpatient rehabilitation: 3,765,190 • Indirect costs: 17,750,554 ○ Sick leave: 17,750,554 Oropharynx: 99,404,580 • Direct costs: 83,029,501 ○ Hospitalization: 78,983,397 ○ Inpatient rehabilitation: 4,046,104 • Indirect costs: 16,375,079 ○ Sick leave: 16,375,079 |

(*Continued*)

**Table 5.** (Continued)

| Study | Year of cost | Country | Currency | Cancer anatomical sites considered for cost | Cost per patient (95% CI) | Cost per patient by clinical stage | Other measures of cost (95% CI) |
|---|---|---|---|---|---|---|---|
| **Longitudinal studies** | | | | | | | |
| 23. Keeping, 2018 [21] | 2011 | England | Pounds (GBP) | Lip; base of tongue; other and unspecified parts of tongue; gum; floor of mouth; palate; oral cavity; other and unspecified parts of mouth; tonsil; oropharynx; larynx | Not applicable | Not applicable | Total cost– 5 years: Oral cavity: 98,330,746 • Inpatient: 94,876,001 • Outpatient: 3,454,745 Oropharynx: 114,619,227 • Inpatient: 108,738,446 • Outpatient: 5,880,781 |
| 24. Milani, 2021 [22] | 2018 | Brazil | International Dollar (I$) | Lip, base of tongue, other and unspecified parts of tongue, gum floor of mouth, palate, other and unspecified parts of mouth, tonsil, and oropharynx | * Average per hospitalization: 1,376 • Lip: 655 • Oral cavity: 1,640 • Oropharynx: 1,338 | Not applicable | Total cost– 9 years: 495.6 million • Inpatient: 244.0 million ○ Professional: 117.9 million ○ Hospital service: 92.5 million ○ Intensive care unit:33.6 million • Outpatient: 251.6 million Lip: 22.7 million • Inpatient: 18.1 million ○ Professional: 9.2 million ○ Hospital service: 7.9 million ○ Intensive care unit: 1.0 million • Outpatient: 4.6 million Oral cavity: 257.1 million • Inpatient: 139.1 million ○ Professional: 69.6 million ○ Hospital service: 55.2 million ○ Intensive care unit: 14.3 million • Outpatient: 118.0 million Oropharynx: 215.8 million • Inpatient: 86.8 million ○ Professional: 39.1 million ○ Hospital service: 29.4 million ○ Intensive care unit: 18.3 million • Outpatient: 129.0 million |

US, United States; GDP, Gross Domestic Product; 95% CI, 95% Confidence Interval; BT group, Brachytherapy group; S group, Surgery group; EBRT group, External beam radiotherapy group; IR, interquartile ranges 25%-75%; NR, Non reported; NA, Not applicable.

* Average cost estimate per event and total cost calculated by the authors of the present study according to the data reported for both sees.

Governments and health insurance providers are supposed to be the organizations supplying support to the population in order to face the high cost of chronic diseases. Nevertheless, oral cancer has a 90% chance of being cured, if detected early [42, 43].

The development of effective public policies is crucial for reducing these health expenditures. Oral cancer is confirmed as a public health problem and was a concern of at least 17 countries on 4 continents, based on the studies included in this review.

**Table 6.  Burden of cost of lip, oral cavity, and oropharynx cancers according to Gross Domestic Product per capita.**

| Study | Country (currency) | PPP* conversion factor | GDP** per capita (PPP 2019) | Components | Lip | Oral cavity | Oropharynx |
|---|---|---|---|---|---|---|---|
| 1. Kim, 2011 [23] | UK (GBP) | | | Cost per patient | 5,790 | 25,311 | --- |
| | | 0.68 | 46,659 | **% GDP *per capita*** | **18.3%** | **79.8%** | --- |
| 2. Polesel, 2019 [26] | Italy (EUR) | | | Cost per patient | --- | 18,462 | 24,253 |
| | | 0.67 | 42,492 | **% GDP *per capita*** | --- | **64.9%** | **85.2%** |
| 3. van Agthoven, 2001 [35] | The Netherlands (EUR) | | | Cost per patient | --- | 35,541 | 35,642 |
| | | 0.78 | 56,935 | **% GDP *per capita*** | --- | **79.8%** | **80.3%** |
| 4. Lairson, 2017 [41] | EUA (USD) | | | Cost per patient | --- | --- | 134,454 |
| | | 1.00 | 62,530 | **% GDP *per capita*** | --- | --- | **215.0%** |

*PPP: Purchasing Power Parity.

**GDP: Gross Domestic Product.

The main characteristics that qualify a COI study are expressed in its methodological definition. These include, among other aspects, the epidemiological approach, costing method and data collection. Incidence-based COI studies should include both direct and indirect costs throughout the life course to outcome. Prevalence-based COIs also include direct and indirect costs over a given period from any stage of the disease. For an acute illness, these two approaches would estimate similar costs. However, for a chronic disease, such as oral cancer, longitudinal incidence-based studies would provide more accurate estimates of the costs of this disease overt time. Considering the costing method for identifying and measuring resources, the COI approach can be micro (bottom up) or macro costing (top down). Using the micro-costing method, costing components and items are measured at the most detailed level possible, with estimated costs per individual, and the selection of a representative sample is recommended to allow external validity or generalizability of the results to a broader population. In macro costing, the total aggregate cost is divided by the number of individuals and can be expressed as an average value. Generally, COI studies that use micro-costing are more accurate, but less generalizable. Regarding data collection, retrospective studies represent a challenge because the data are secondary, generally intended for other purposes (epidemiological or surveillance) and may not be sufficient for a COI study. Most of the studies included in this systematic review did not meet all the items of the instrument used for quality assessment.

Although the economic burden of oral cancer was substantial, this systematic review showed that the costs may be underestimated, and only one [19] of the 24 studies considered all components of cost-of-illness simultaneously. In addition, from the six studies that analyzed indirect costs [18, 19, 28–31], only three studies [19, 29, 31] included costs of early death related to the disease, which is one of the most expensive items for society [44]. Further longitudinal studies with higher quality are needed, not only methodologically, but in their data analysis and reporting of results. These studies should include, not only direct medical costs, but also direct non-medical and indirect costs, so that more accurate estimates can contribute to cost evaluation of health promotion and disease management programs.

The wide heterogeneity of COI studies was identified in both the aspects related to disease characterization and those related to economic issues. Regarding the disease, the main sources of heterogeneity were the characteristics of the samples; the lack of standardization in the definition of the clinical stage of the disease, and the different sets of head and neck tumors studied. The heterogeneity related to economic issues of the studies were found in their design,

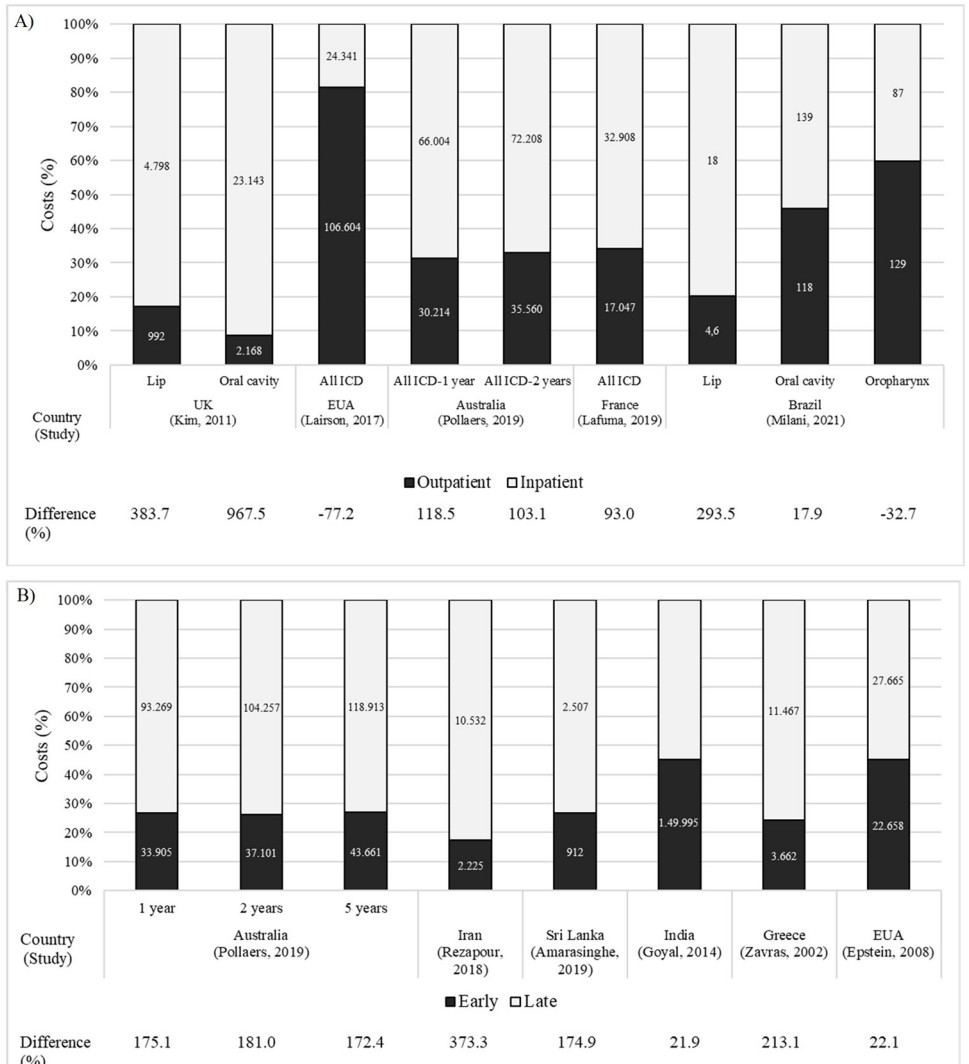

**Fig 2.** Oral cancer burden of cost and difference of costs (%) according to types of patient care (A) and clinical stage of the disease (B). Difference (%) = [(inpatient cost–outpatient cost)/outpatient cost x 100]. Currency: Kim,2011: Pounds; Lairson, 2017, Rezapour, 2018, Amarasinghe, 2019, Zavras, 2002, Epstein, 2008: US dollars; Pollaers, 2019: Australian dollars; Lafuma, 2019: Euros; Milani, 2021: International dollars (million); Goyal, 2014: rupees.

perspective, time horizon, sources of information, components and costing items, the health system of each country, currency, and reporting of cost results. The World Health Organization (WHO) recommendation is that the results of COI studies be reported in international dollars according to the PPP, to better support country-to-country comparisons of costs [45]. The development of protocols for the cost evaluation of oral cancer should be encouraged, as it has been by the Pan American Health Organization (PAHO) with the protocol for calculating the cost of hospital infections [46], since these analyses are complex and depend on the objectives of the studies. Protocols may contribute to the reduction of heterogeneity, favoring the comparison between different regions and health systems, in order to obtain a more accurate calculation of oral cancer cost.

In general, inpatient costs are higher than outpatient costs. However, this depends on the provision of health resources in each healthcare unit of the health system in each country. For

example, in the USA [41], outpatient costs were higher than inpatient costs because most patients were treated with radiotherapy in outpatient care, which is one of the most expensive treatments for oral cancer management.

The costs of oral cancer in advanced clinical staging were higher than those at early stages, which occurred regardless of the heterogeneous characteristics of the studies. Most cases of oral cancer have been diagnosed in advanced staging for almost two decades [47] which in addition to compromising patient survival, determines high-cost treatments and suggests flaws in policies to promote preventive measures/strategies and early detection and diagnosis. This reinforces the importance of public policies that prioritize actions in the context of primary care, including health education for the population, qualification of professionals for the early detection of the disease, and the monitoring of the population at risk through opportunistic screening [7, 48].

The main limitation of this review was the difficulty of finding average cost results per patient from cancer sites, defined here as oral cancer (ICD-10 C01-C06, C09, and C10). These difficulties are possibly associated with the presentation of study results as aggregate costs of head and neck cancer and, also as a result of the absence of an international standardization defining which anatomical sites should characterize oral cancer. The heterogeneity of studies in other aspects of the disease characteristics, method, and economic issues may also have impacted on our findings, which did not allow a meta-analysis.

Decision makers increasingly require economic evidence to inform health policies [49, 50] and systematic reviews of economic evaluations (COI and cost-effectiveness) have grown accordingly [51–55]. This study provides a comprehensive and critical overview of the COI analyses conducted around the world, which highlights the magnitude of the financial impact of oral cancer on societal or public health expenditure. This evidence can contribute to priority setting, particularly in the context of scarce resources. Our results can also be used by several other key stakeholders, such as international organizations (WHO and World Bank), health insurance companies, and health providers (health facilities and health workers).

This systematic review can also provide relevant insights for the health technology assessment field, particularly for economic evaluation studies. COI studies represent the first step towards complete economic analysis (e.g., cost-effectiveness analysis) and can also support budgetary impact analysis, by identifying, measuring, and valuing costs related to a specific disease or health condition [56].

## Conclusion

This systematic review shows that the economic burden of oral cancer is substantial and underestimated. The cost of LC, OCC, and OPC reach an average of 18%, 75%, and 127% of GDP per capita, respectively, in some western countries. Further high-quality COI studies are needed, especially with robust methodological design and those that include, in addition to direct medical costs, the direct non-medical and indirect costs. Standardization of the terminology of the types of cancer and clarity in reporting the sources of cost information are crucial to consider in the COI studies. Also, if COI studies present international dollars as the unit price to reflect the economic cost of goods, and allow inter-country comparison of costs, this could support policy makers to identify major cost drivers of oral cancer and to make decisions regarding a more effective public policy for the prevention of oral cancer.

## Supporting information

**S1 Checklist. PRISMA 2020 checklist.**
(PDF)

## Author Contributions

**Conceptualization:** Rejane Faria Ribeiro-Rotta, Eduardo Antônio Rosa, Danielle Masterson, Everton Nunes da Silva, Ana Laura de Sene Amâncio Zara.

**Data curation:** Rejane Faria Ribeiro-Rotta, Eduardo Antônio Rosa, Ana Laura de Sene Amâncio Zara.

**Formal analysis:** Ana Laura de Sene Amâncio Zara.

**Funding acquisition:** Rejane Faria Ribeiro-Rotta.

**Investigation:** Rejane Faria Ribeiro-Rotta, Eduardo Antônio Rosa, Vanessa Milani, Nadielle Rodrigues Dias, Everton Nunes da Silva, Ana Laura de Sene Amâncio Zara.

**Methodology:** Rejane Faria Ribeiro-Rotta, Eduardo Antônio Rosa, Danielle Masterson, Everton Nunes da Silva, Ana Laura de Sene Amâncio Zara.

**Project administration:** Rejane Faria Ribeiro-Rotta, Eduardo Antônio Rosa, Ana Laura de Sene Amâncio Zara.

**Resources:** Rejane Faria Ribeiro-Rotta.

**Supervision:** Danielle Masterson, Everton Nunes da Silva.

**Validation:** Rejane Faria Ribeiro-Rotta, Danielle Masterson, Everton Nunes da Silva, Ana Laura de Sene Amâncio Zara.

**Visualization:** Rejane Faria Ribeiro-Rotta, Eduardo Antônio Rosa, Ana Laura de Sene Amâncio Zara.

**Writing – original draft:** Rejane Faria Ribeiro-Rotta, Eduardo Antônio Rosa, Ana Laura de Sene Amâncio Zara.

**Writing – review & editing:** Rejane Faria Ribeiro-Rotta, Eduardo Antônio Rosa, Vanessa Milani, Nadielle Rodrigues Dias, Danielle Masterson, Everton Nunes da Silva, Ana Laura de Sene Amâncio Zara.

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
