## [Decision Letter · Decision Letter 0]

22 Nov 2021

PONE-D-21-31795The cost of oral cancer: a systematic reviewPLOS ONE

Dear Dr. ROTTA,

Thank you for submitting your manuscript to PLOS ONE. After careful consideration, we feel that it has merit but does not fully meet PLOS ONE’s publication criteria as it currently stands. Therefore, we invite you to submit a revised version of the manuscript that addresses the points raised during the review process.

We look forward to receiving your revised manuscript.

Kind regards,

Antoine Eskander, MD, ScM, FRCSC

Academic Editor

PLOS ONE

Additional Editor Comments (if provided):

Consider the major comments submitted by both reviewers in your resubmission.

Reviewers' comments:

Reviewer's Responses to Questions

**Comments to the Author**

1. Is the manuscript technically sound, and do the data support the conclusions?

Reviewer #1: Yes

Reviewer #2: Yes

2. Has the statistical analysis been performed appropriately and rigorously? 

Reviewer #1: N/A

Reviewer #2: N/A

3. Have the authors made all data underlying the findings in their manuscript fully available?

Reviewer #1: No

Reviewer #2: Yes

4. Is the manuscript presented in an intelligible fashion and written in standard English?

Reviewer #1: Yes

Reviewer #2: Yes

5. Review Comments to the Author

Reviewer #1: I thank the authors for their submission. This is a nice systematic review of the economic burden of oral cancer. From a methodologic perspective, this work is rigorous and adheres to many best practices including a completed PRISMA checklist as well as an a priori protocol, available and registered on PROSPERO.

My main reservation is with the concept of cost of illness studies more broadly. Cost of illness studies are frequently criticized for not being grounded in welfare theory. They are descriptive and felt to have limited utility in informing resource allocation, particularly in comparison to other forms health economic evaluation (eg. cost-effectiveness analyses). Nonetheless, by comprehensively outlining direct and indirect cost components across various countries, I do think this review may assist with future cost-effectiveness studies in oral cancer. The authors touch on some of these issues briefly in the limitations section, though I do think the limitations of COI should be discussed more explicitly.

The quality of included studies was poor. On pages 31 and 32 the authors very briefly outline what constitutes a high-quality COI study in oral cancer. This should be further elaborated on. I think many of these concepts might not be well known to the head and neck oncology audience. A concise but explicit discussion of prevalence vs. incidence approaches, top-down vs bottom up, prospective vs. retrospective etc. might be helpful to the reader.

Minor Comments:

-I appreciate the explicit use of ‘PICO’ in the development of the research question, though do not think that an abbreviation for population (P) , exposure (E), and outcome (O) are needed in the abstract.

-Could the authors explain why other health economic evaluation studies (CEA, CUA etc…) were explicitly excluded? Presumably some of these studies would have tabulated the direct and indirect costs of oral cancer and could potentially have been included?

-For the Larg and Moss checklist, how did the authors define the threshold of >80% points as ‘high quality’ and 50-79% as ‘medium quality’? Is that an accepted approach?

-Do the authors have any statistical measures of agreement during the article screening phase?

-In the results section “€35,642 in a mathematical model estimated for 10 years”. Could you briefly explain what the ‘mathematical model’ was?

-Table 2 – How did the authors categorize the study as a ‘cost-analysis’ vs ‘cost-of-illness’ study? Could the authors clarify either in the body of the manuscript or as a footnote in the table?

-Additional files 1 and 2 appear to be missing, though I was able to piece together the search strategy and MESH terms from the published protocol. The only additional file included is the completed PRISMA checklist.

Reviewer #2: Thank you for inviting me to review this article. This systematic review study identified 24 studies (2001-2021) published around the world on the cost of oral cancer. Both direct and indirect costs were considered. Grey literature and non-English papers were also searched. The included studies mostly followed the standard cost-of-illness approach and adopted a provider (hospital) perspective. Direct medical costs were examined by most studies, while direct non-medical costs and indirect costs associated with premature mortality and work absence were seldom assessed. Four studies have estimated burden of illness using cost per patient over per-capita GDP. These studies found the cost burden of lip, oral cavity, and oropharynx cancers to amount to 18.3%, 74.8%, and 126.8% of the per-capita GDP of various developed nations.

This is a well-conducted systematic review with meaningful economic outputs to inform resource allocation in cancer care. I have a few major concerns about the review methodology and other minor suggestions that the authors may want to consider in their revisions.

Major comments:

1. It is unclear how the cost items in Table 4 and lines 151-154 were defined. If the authors want to survey all types of costs associated with oral cancer (i.e., in order to understand the economic burden of oral cancer), why use an a priorly defined list of cost items to deliberately limit the scope of cost? Are these cost items derived from an established costing framework for oral cancers from the literature? If not, the authors should include a section that outlines how they determined and refined a costing framework and cost components which by itself should be a contribution of this review.

The authors might want to consult this systematic review on the methodology of defining a costing framework: Clarke K, Klarenbach S, Vlaicu S et al. The direct and indirect economic costs incurred by living kidney donors—a systematic review. Doi: 10.1093/ndt/gfl069

In particular, the authors only considered two types of indirect costs (work absenteeism and premature mortality). What about the indirect loss due to disability or the extra expense of senior care/childcare due to the patient being hospitalized and unable to provide such care? All these questions beg for a costing framework that exhaustively identifies and categorizes all potential cost components of oral cancer before the literature search.

2. The authors can enhance this study significantly by formally defining the key elements of an economic analysis before the literature search. For example, in what way does a cost-of-illness study differ from a cost study (Table 2)? Why is Patterson (2020) a cost-of-illness study (Table 2) if it has only assessed the indirect costs of premature mortality resulted from oral cancers (Table 4)? Another key concept that begs for clarification is perspective (Table 2). I find it unconventional to categorize “perspectives” into hospital, government, payer, and society. It appears the authors are mixing up two concepts: 1) the provider of care or the setting in which the care occurs, such as hospital; 2) the payer who affords the cost of care, such as the government (public payer) or the society. I would make 2 columns in Table 2 to distinguish these entities and add a formal definition for perspective in the methods section. Furthermore, if Milani (2021) used a government (public payer) perspective, it should not have included any direct non-medical costs (unless the government reimburses for these costs). Same with Han (2010) and Amarasinghe (2019).

3. It is absolutely essential that a common currency & year (such as 2019 USD) is used throughout the manuscript (and Table 5) to make between-study comparisons and summarization of costs meaningful. Could the authors convert all currencies (and explain how it is converted) to a common currency & year and present both the initial and converted currency? One way to do this is to first inflate the initial currency to 2019 constant values using the national annual CPI and then adjusted to 2019 USD (or another currency) using PPP from OECD. A minor point is to please remove the currency symbols and replace them with the ISO codes for currencies.

Minor comments:

Methods

Lines 98-99: why explicitly excluding cost-effectiveness/cost-utility studies since they also assessed cost? Studies might not examine costs as their primary objective but nonetheless present useful data. Perhaps the authors want to state that they only included studies whose primary objective was to assess the cost of oral cancer.

Results

In Table 1, the authors might want to report more information of cohort from each study (age, sex composition etc) beyond the size of cohort. Difference of the study cohort contributes to the high between-study heterogeneity as well.

Page 19 (Table 4): please revise Table 4 to reflect two major categories of costs (direct vs. indirect costs), and under direct costs the authors can further define direct medical costs and direct non-medical costs.

Page 30 (Table 6): since only 4 studies reported the economic burden of oral cancer as a percentage of per-capita GDP, and all studies were based in developed western countries, I would not state these results as a general finding in the abstract.

Furthermore, only one of these 4 studies (Kim 2020) has examined all cost components as defined in Table 4. Then, why is the cost-per-patient estimate from the remaining 3 studies reliable if the authors have already identified them to represent an underestimation?

Alternatively, the authors might want to use the estimate of Rezapour (2018) to manually compute the burden of disease as a percentage of total cost in per-capita GDP.

---

## [Author Response · Author response to Decision Letter 0]

7 Feb 2022

See attached file: Response to reviewers

---

## [Decision Letter · Decision Letter 1]

21 Mar 2022

The cost of oral cancer: a systematic review

PONE-D-21-31795R1

We’re pleased to inform you that your manuscript has been judged scientifically suitable for publication and will be formally accepted for publication once it meets all outstanding technical requirements.

Kind regards,

Antoine Eskander, MD, ScM, FRCSC

Academic Editor

PLOS ONE

Additional Editor Comments (optional):

Thank you for addressing all concerns.

Reviewers' comments:

Reviewer's Responses to Questions

**Comments to the Author**

1. If the authors have adequately addressed your comments raised in a previous round of review and you feel that this manuscript is now acceptable for publication, you may indicate that here to bypass the “Comments to the Author” section, enter your conflict of interest statement in the “Confidential to Editor” section, and submit your "Accept" recommendation.

Reviewer #1: (No Response)

Reviewer #2: All comments have been addressed

2. Is the manuscript technically sound, and do the data support the conclusions?

Reviewer #1: Yes

Reviewer #2: Yes

3. Has the statistical analysis been performed appropriately and rigorously? 

Reviewer #1: Yes

Reviewer #2: Yes

4. Have the authors made all data underlying the findings in their manuscript fully available?

Reviewer #1: Yes

Reviewer #2: Yes

5. Is the manuscript presented in an intelligible fashion and written in standard English?

Reviewer #1: Yes

Reviewer #2: Yes

6. Review Comments to the Author

Reviewer #1: (No Response)

Reviewer #2: (No Response)

7. PLOS authors have the option to publish the peer review history of their article (what does this mean?). If published, this will include your full peer review and any attached files.

Reviewer #1: No

Reviewer #2: **Yes: **Rui Fu

---

## [Editor Report · Acceptance letter]

30 Mar 2022

PONE-D-21-31795R1 

The cost of oral cancer: a systematic review 

Dear Dr. ROTTA:

I'm pleased to inform you that your manuscript has been deemed suitable for publication in PLOS ONE. Congratulations! Your manuscript is now with our production department. 

Kind regards, 

on behalf of

Dr. Antoine Eskander 

Academic Editor

PLOS ONE